# Measuring multi-dimensional disparity index: A case of Nepal

**Prakash C. Bhattarai** [1]*, **Milan Shrestha**[2], **Prakash Kumar Paudel**[3]

**1** Kathmandu University-School of Education, Hattiban, Lalitpur, Nepal, **2** Tribhuvan University-Graduate School of Education, Kritipur, Kathmandu, Nepal, **3** Department of Development Education, Kathmandu University-School of Education, Hattiban, Lalitpur, Nepal

* prakash@kusoed.edu.np

**Data Availability Statement:** All relevant data are within the manuscript and its Supporting information files.

**Funding:** The authors received no specific funding for this work.

## Abstract

This study introduces Multi-dimensional Disparity Index (MDI) to measure multi-form of disparity in different level of governments referencing Nepal. The measurement scale of MDI was developed by adopting Santos and Alkire's (2011) approaches. A wide range of thematic experts was consulted, employing the Semi-Delphi approach to determine its dimensions and indicators. The MDI in this study includes six dimensions and 34 indicators composited with dimension-wise indices like Economy Disparity Index [ECODI], Educational Disparity Index [EDUDI], Health Disparity Index [HDI], Geography and Climatic Vulnerability Index [GCVI], Living Standard Disparity Index [LSDI], and Demography Disparity Index [DDI]. Overall, the study revealed the extent of multi-dimensional disparity across three tiers of government in Nepal. More specifically, Nepal scored 0.388 MDI value. Karnali and Bagmati provinces are accounted as the highest and least deprived. This study contributes essential knowledge, particularly in exploring the dimensions and their indicators and develops an approach to measure multi-dimensional disparities. Most existing approaches for assessing disparities are mono-dimensional and measure the disparities in a single aspect. In this context, MDI provides a broader approach to consider multiple dimensions and measures multiple aspects in a country like Nepal, where disparity manifests at multiple levels.

## Introduction

Nepal has been a unitary state for a long, particularly after King Prithvi Narayan Shah, the first monarch of unified Nepal, proclaimed a new unified Nepal in the 18th century. King Shah and his many descendants engaged in the war and put much effort into balancing power across history; thus, the socioeconomic aspect of the country became a neglected agenda for a long [1]. Subsequently, the Rana Regime, followed by the Panchayat system, also dims to harness inclusive development practices in Nepal. As a result, the disparity is perpetuated at multiple levels among Nepali citizens. The inequality among the people also increased dissatisfaction among the citizens, resulting in consequences such as ten years-long civil war (from Feb 13, 1996, to Nov 21, 2006) [2, 3]. Nevertheless, the civil war paved the way for a decentralized governance system. Nepal was declared a federal state in 2015, restructuring the unitary system into 753 local and seven provincial governments [4]. One of the aims of this nation's restructuring into

**Competing interests:** The authors have declared that no competing interests exist.

the new federal system is to decentralize the unitary governance system and ensure equitable distribution of resources to all units.

The equitable distribution of resources and access to services among people also becomes a priority in the national context. Nepal, in this case, transits towards the inclusive governance system that aims to distribute resources equitably among the people. However, the studies conducted in Nepal [5–9], show disparities are still prevalent among the citizens, particularly in terms of poverty, gender, and access to resources. Furthermore, these disparities are multi-dimensional and are deeply rooted in socioeconomic and geographical diversity [8].

The disparities prevail across the provinces in Nepal in multiple aspects. For instance, Gandaki Province (0.513) is ranked better than Karnali (0.412) and Sudhur-Paschim (0.415) provinces in Human Development Index (HDI) value [10]. The two provinces also accounted least, Sudur-Paschim (0.676) and Karnali (0.680), in Gender Inequality Index (GII) [11]. The Multi-dimensional Poverty Index (MPI) score of Bagmati province (0.051) is lower than other provinces like Karnali (0.230) and Madhesh (0.217) respectively [12]. Additionally, Bagmati province relatively has better indicators such as literacy rate (74.8% compared to 49.5% of Karnali) [13], life expectancy 70 years in comparison to 67 years of Sudur-Paschim [14]. The evidence shows that the disparity in Nepal exists in multi-dimensions.

The multi-dimensional disparity is the state of variances that denotes the gap [15] in distributing resources and services between subjects in multiple sectors. The multi-dimensional disparity is also associated with the human capability approach [16] that emphasizes the multi-dimensional components of wellness and the social arrangement of people [17–20]. Well-being and similar social arrangement refer to the absence of disparity among people. In this case, the disparity has more valuable functions than resources [17, 19, 21]. This consideration also applies to explaining capability deprivation which assesses differences among people [22]. Developing countries like Nepal have many forms of disparity associated with existing social, economic, political, and geographical structures [23]. There are many disparity indexes such as; GII, Gender Disparity Index (GDI), Health Disparity Index (HDI), and Educational Disparity Index (EDI); however, these indexes only explain the disparity in a single dimension. In this context, developing this Multi-dimensional Disparity Index (MDI) gives a more robust framework to measure multi-dimensional forms of disparity in the countries where disparity perpetuates in various forms. Furthermore, in this study, MDI provided us with an approach for assessing the multi-form of socio-economic disparities.

We employed MDI to explain the extent of the overall disparity by incorporating a wide range of areas of six different dimensions of disparities. So, we set our assumptions of disparity based on indicators such as; poverty by Multi-dimensional Poverty Index_MPI [12, 24, 25], gender disparity by Gender Inequality Index_GII [26], health disparity through Health Disparities Index_HDI [27], education inequality by Gini Coefficients of education [28], and income via Gini Index [29]. This study, taking diverse forms of disparities into account, intends to assess the Multi-dimensional Disparity Index of Nepal in all seven provinces. In addition, this study also aims to elucidate the six dimensions of MDI as the Economy Disparity Index (ECODI), Educational Disparity Index (EDUDI), Health Disparity Index (HDI), Geography and Climatic Vulnerability Index (GCVI), Living Standard Disparity Index (LSDI), and Demography Disparity Index (DDI), respectively of Nepal and its Provinces. Furthermore, the distribution of local governments basis on the level of MDI values within the provinces of Nepal is portrayed in this study.

## Dimensions of MDI

MDI is the overall form of inequality index that incorporates multiple dimensions of MPI, HDI, Living Standard Index (LSI), Health Disparity Index (HDI), and Educational Gini Index (EGI). Moreover, the global MPI consists of three dimensions as health, education, and living standard [24, 30–32]. Similarly, HDI also assesses three dimensions: long and healthy life, knowledge, and a decent standard of living [33]. More specifically, "long and healthy life" is measured by life expectancy at birth, "knowledge" is a composite form of "expected years of schooling" and "years of schooling," and "decent standard of living" is assessed via Gross National Income (GNI) per capita [34]. Exceeding this, the South African Index of Multiple Deprivation (SAIMD) includes six dimensions: education, health, living standard, housing, economics, and demographic [35], analogous to the MDI. However, MDI merged housing to the living standard and formed a single dimension known as the living standard. The MDI further includes the geography and climatic vulnerability dimension, and it is a composite of six dimensions economy, education, health, geography and climatic vulnerabilities, living standard, and demography.

The economy is the first dimension of MDI; we denote it by "financial security and dignified work," the fourth domain of the Multi-dimensional Inequality Framework (MIF). It is calculated including "Inequality in achieving financial independence and security, dignified and fair work, and recognition of unpaid work and care" [22]. The first dimension as economy is also analogous to the economic dimension of SAIMD, which assesses the lack of employment, income, and dependency ratio of youth and the elderly [35]. In premises of these dimensions, in this study, the economic dimension is a composite of the bank interface, availability of regular air transport, type of local government, transportation accessibility, telephone and internet accessibility, electricity accessibility, forest area, industry establishment, and active population engaged ratio.

We consider education as the second dimension of MDI. Alike in this study, provincial multiple deprivation indices in South Africa included the education dimension accounting the number of schooling years as its indicator [36]. Similarly, SAIMD also examined education with indicators such as; primary and secondary years of schooling and school attendance [35]. In contrast, MPI included adult literacy and school enrollment as its indicators instead of secondary years of schooling and child school attendance in the context of Nepal [24]. In this study, we considered the availability of campus/school, a dropout from 1–5 classes, school-student ratio, Gross Enrollment Ratio (GER) of secondary level, and illiteracy rate in the education dimension.

We enlist health as the third dimension of MDI, similar to the SAMID [35, 36]. In SAMID, health comprises child mortality, adult mortality, and parenthood survival. Furthermore, "years of potential life lost" is also included to assess deprivation of provinces in SAMID [36]. We go beyond this literature and incorporate six indicators; health facility, life expectancy, neonatal deaths, maternal deaths, fully immunized children, and child nutrition (under two years). Likewise, geographical and climatic factors are also considered to affect the equal distribution of resources and opportunities in Nepal [37]. In this case, geographical and climatic vulnerabilities become the fourth dimension which includes five indicators; disaster-affected household, annual precipitation trend, overall vulnerability, distance from district headquarters, and geographical regions.

Living standard is the fifth dimension of this study and it is also used in calculating MPI [38]. Unlike indicators in MPI such as; availability of cooking fuel, electricity, improved sanitation, safe drinking water, the house itself and other assets for convenient human life, we set ownership of houses and vehicles, facility of electricity, and drinking water as the indicators of living standard.

Finally, demography is the sixth dimension. Moreover, we use household size, gender differences, population density, disability ratio, and dependent population as composite indicators. Overall, these six dimensions with their indicators were measured to derive the MDI score in all provinces of Nepal. Along with MDI, we also assessed each disparity index like ECODI, EDUDI, HDI, GCVI, LSDI, and DDI for all these dimensions.

## Methodology

This research assessed the level of MDI and its dimensions considering Nepal and its provinces. Nepal is a land-locked country in the middle of Asia, particularly between two big economic countries; China and India. It is the melting pot between China and India in terms of socio-economic aspects [39]. So, it fairly represents the socio-economical features and MDI of both countries, even the majority of Asia.

### Brief descriptions of the subject of this study

There are seven provinces: Koshi, Madhesh, Bagmati, Gandaki, Lumbini, Karnali, and Sudhur-Paschim in Nepal [4, 40] which are referring as the subject of this study (For detail, see the S1 Fig, S1 Table in S1 Appendix). Koshi province is located in the eastern part, Madhesh is in the southeast, Bagmati and Gandaki are in the central region, Lumbini is in the southwest, Karnali is in the northwest, and Sudhur-Pashchim is in the far western part of Nepal. These provinces cover diverse geographical regions (e.g., Mountain, Hill, and Tarai plain) except Karnali, which only consists of Mountainous and hilly regions, and Madhesh only occupies a southern plain area of Nepal. Additionally, Madesh province is the smallest but has the highest population density, while Karnali is Nepal's largest but least populated province [41]. Moreover, the distribution of local government also differs across these provinces despite their areas and population. Koshi has 137 local governments, followed by Madhesh (136), Bagmati (119), Gandaki (85), Lumbini (109), Karnali (79), and Sudhur-Paschim (88) [4] respectively.

### Sources of data

This study calculated the MDI using secondary data obtained from various government sources. In line with this, Table 1 provides the data sources for computing the overall MDI score.

CBS was the primary data source in this study as it is the largest government authority in charge of conducting research. However, we also obtained data from other government

**Table 1. Dimensions of MDI and its sources.**

| S. No. | Dimensions of MDI | Sources |
|---|---|---|
| 1 | Economy | NPC & Central Bureau of Statistics (CBS), 2020 [42]; CBS, 2022 [41]; Ministry of Federal Affairs and General Administration (MOFAGA), 2019 [43]; Department of Forest Research and Survey (DOFRS), 2018 [44] |
| 2 | Health | Department of Health Services (DOHS), 2022 [14]; CBS, 2022 [41] |
| 3 | Education | Centre for Education and Human Resource Development (CEHRD), 2022 [45]; CBS, 2022 [41] |
| 4 | Geography and Climatic Vulnerability | CBS, 2019 [46]; Nepal Disaster Risk Reduction Portal (NDRRP), 2022 [47]; MOFAGA, 2019 [43]; Department of Hydrology and Meteorology (DOHM), 2017 [48] |
| 5 | Living standards | CBS,2022 [41] |
| 6 | Demography | CBS, 2022 [41] |

authorities in specific areas, such as DOHS, CEHRD, NPC, MOFAGA, DOFRS, NDRRP, and DOHM. These sources of data were published from 2017 to 2022 (For detail, see the S2 Table in S1 Appendix). The composite of these six dimensions is the index score of multi-dimensional disparities of local government of respective province.

## Dimensions and indicators

The six dimensions of MDI in this study consist of 34 indicators which are presented in Table 2.

These dimensions and their indicators of MDI were identified by employing the Semi-Delphi approach. For this, first, we identified 18 experts working in government, funding agencies, academia, and social sectors. We had several consultative interactions with these experts individually and in a group. In the interaction, commonly, we explained the purpose of our study, MDI and its possible dimensions with several probing questions. These interactions helped to construct and validate the proposed dimensions and their indicators. Then, we compiled all the dimensions and their indicators explored in the consultative interaction and sent them back to the consulted experts for the rating with five responses: highly important, important, moderately important, not important, and highly unimportant [49]. After getting responses from all experts, we set 70% response as the level of consensus for all items and thus established consensus among experts on dimensions and indicators of MDI. Overall, 34 items were retained within six dimensions, including the economy (9 items), education (5 items), health (6 items), geography and climatic vulnerabilities (5 items), living standards (4 items), and demography (5 items), respectively.

## Data analysis procedures

We analyzed data in four steps; indexing the indicators, assigning the weights and deprivation scores, identifying the disparity cut-offs and multi-dimensional disparity (Ci) of local government, and finally calculating the MDI of provinces and Nepal.

**Indexing the indicators.** We employed the mini-max indexing of Mustaffa and Yusof approach for normalizing the values between 0 to 1 [50]. This approach makes all indicators in

**Table 2. Dimensions and their indicators of MDI.**

| S. No. | Dimensions | Indicators and its Code |
|---|---|---|
| 1 | Economy | Bank interface (ECO1), availability of regular air transport (ECO2), type of municipality (ECO3), Transportation accessibility (ECO4), Telephone and internet accessibility (ECO5), electricity accessibility (ECO6), forest area (ECO7), industry establishment (ECO8), active population engaged ratio (ECO9) |
| 2 | Education | Availability of campus/school (EDU1), Dropout from1-5 class (EDU2), school student ratio (EDU3), GER of SS level (EDU4), Illiteracy rate (EDU5) |
| 3 | Health | Health facility (H1), Life expectancy (H2), Neonatal deaths (H3), Maternal deaths (H4), Fully Immunized Children (H5), Nutrition of Child under 2 years (H6) |
| 4 | Geography and Climatic Vulnerability | Disaster affected household (GCV1), Annual precipitation trend (GCV2), Overall vulnerability (GCV3), Distance from district headquarter (GCV4), Geography (GCV5) |
| 5 | Living standards | Electricity (LS1), Tap water (LS2), ownership of house (LS3), Having vehicle (LS4) |
| 6 | Demography | Sex ratio (D1), Household size (D2), Population density (D3), Disability ratio (D4), Dependent population (D5) |

similar forms of units and derives the index values of each indicator. The index values are obtained by comparing the sum of actual (Xi) and maximum values (Xmax) and subtracted by its minimum (Xmin) values as in Eq 1.

$$\text{Standardized index} = (X_i - X_{min})/(X_{max} - X_{min}) \tag{1}$$

**Assigning the weights and deprivation scores.** We set the hypothesis that all index dimensions assign an equal weight and every indicator within that dimension also carries an identical weight [51, 52]. Each dimension of MDI was allocated one-sixth of the total weight. Furthermore, each indicator's weight (i) within the dimension is assigned as economy, education, health, geography and climatic vulnerability, living standards, and demography, respectively. Overall, the weights of indicators of six dimensions summed up to 1. These weights of indicators were obtained based on the model of Santos and Alkire [52] [eg. 12, 25, 35] in Eq 2.

$$W_1 = \sum_{i=0-1}^{d} w_i \tag{2}$$

Where,

$W_1$ = weight of indicator first
$W_i$ = weight attached to indicator i
d = no. of dimension
i = weight (0 to 1), where the value is between 0 and 1

In Eq 2, the weight (W) of the indicators contributes to obtaining the indicators' deprivation score (i). The deprivation score of each indicator refers to the degree of disparity where a value near '0' indicates a low disparity and '1' considers a high disparity [52].

**Identifying the disparity cut-offs and multi-dimensional disparity (Ci) of local governments.** We used the Alkire Foster measurement framework as a disparity-cutoff strategy [52]. Firstly, we applied a deprivation cut-off (dimension-specific cut-off) to each dimension. Then, we ranged below or above its cut-off, considering that each local government in the province is either deprived or not by its achievement of each or overall dimensions. For instance, a local government is deprived when its overall weighted sum of disparity (Ci) meets or exceeds multi-dimensional cut-offs [18]. The weighted sum of disparity (Di) of each dimension refers to the disparity index (e.g., Economy Disparity Index [ECODI], Educational Disparity Index [EDUDI], Health Disparity Index [HDI], Geography and Climatic Vulnerability Index [GCVI], Living Standard Disparity Index [LSDI], and Demography Disparity Index [DDI]). To calculate the provincial level value, first, we calculate the local government's overall dimensional value (Ci), which refers to the local government level's MDI. The local government's multi-dimensional disparity (Ci) was obtained by employing Eq 3.

$$C_i = W_1 I_1 + W_2 I_2 + \ldots\ldots + W_d I_d \tag{3}$$

Where,

$C_i$ = Deprivation indices of a local government (for each dimension and overall)
$W_1$ = weight of indicator first
$I_1$ = 1(if the municipality is deprived) or 0 (if the local government is not deprived) in indicator i

We set 1/6 or 0.166 as the multi-dimensional disparity cut-off for the local government level considering the six dimensions of MDI. Connecting it, a local government holding a **$C_i \leq 0.166$** and **$Ci \geq 0.166$** values were referred to be absent and presence of multi-dimensional disparity respectively. Further, each dimension of the Disparity Index (Di) was obtained by

multiplying its total summation of indicators within a particular dimension by six, as in Eq 4.

$$D_i = (W_1 I_1 + W_2 I_2 + \ldots\ldots + W_d I_d) * 6 \tag{4}$$

**Calculation of MDI of Nepal and its provinces.** After obtaining the multi-dimensional disparity index (Ci) of local governments, researchers calculated the MDI of all the provinces of Nepal. The MDI of provinces is calculated via the multi-dimensional headcount ratio (H) and the Intensity of Deprivation (A) [18, 32] as in Eq 5.

$$MDI = H \times A \tag{5}$$

Where,

MDI = Multi-dimensional deprived indices

H = Multi-dimensional headcount ratio

A = Intensity of deprivation

Then,

$$H = \frac{q}{n} \tag{6}$$

Where,

q = number of population which is multi-dimensionally deprived

n = total population within local/province/Nepal

More specifically, the 'H' refers to the headcount ratio in which the percentage of units considered is deemed to be deprived within 753 local governments of Nepal. It is a ratio between the total numbers of the deprived local governments (q) and the overall sum of the populations of the local government (n) within a given province (in Eq 6). Similarly, 'A' is considered as the intensity of deprivations, which is obtained as in Eq 7.

$$A = \frac{\sum_{i=1}^{n} c_i(k)}{q} \tag{7}$$

Where,

q = number of populations which is multi-dimensionally deprived

n = total population within local/provenience/Nepal

The value of multi-dimensional disparity (Ci) ranged from 0 to 1, where '0'refers to non-deprived and '1' consisted of deprived. The local government that is deemed to be not multi-dimensionally deprived of its population is excluded from calculating the MDI of the province.

After obtaining the MDI of provinces, this study categorized all local governments in six levels of MDI within seven provinces by employing Best's (1977) approach [53] as in Eq 8.

$$= \frac{Higher\ score - Lower\ score}{Number\ of\ Levels} = 0.166 \tag{8}$$

Linking to Eq 8, the higher and lower score was referred to as "1" and "0" respectively. So, the six levels of MDI were computed as the first level (below 0.166), second level (0.167–0.333), third level (0.334–0.500), fourth level (0.501–0.666), fifth level (0.667–0.833), and sixth level (above .834) in the alterations of 0.166 score. Then finally, this study elucidated the frequencies and percentages of local governments within these six levels of MDI in seven provinces of Nepal.

### Ensuring validity

We considered content, criterion, and construct measures to ensure the validity of this study. First, this study incorporated all the items of economy, education, health, geography and climatic vulnerabilities, living standards, and demography as the dimensions of MDI. These six dimensions of MDI covered the optimum indicators relating to MPI [7, 24, 35], like other disparity indexes. It included the stable quality of items based on the several relevant literatures about multi forms of disparity. Second, the criteria validity is proven by comparing the acquired score of indicators with other comparable types of studies conducted in disparity indices [7, 24]. Third, extensive interactions in the Semi-Delphi process with several subject experts enriched our study.

## Results

### Distribution of local governments across its MDI values within provinces

We computed the six levels of MDI regarding local governments within seven provinces of Nepal. While obtaining the levels of MDI via Best's (1977) approaches [54], the fifth and sixth levels of MDI were not found in this study. So, there are only four levels of MDI; first level (below 0.166), second level (0.167–0.333), third level (0.334–0.500), and fourth level (0.501–0.666), respectively, in the variances of 0.166. This difference of 0.166 is also the multi-dimensional disparity cut-off which demarcates the values of MDI as the status of non-deprived and deprived regarding each local government. It means that below the value of 0.166 refers to non-deprived, whereas above 0.166 is a deprived level of MDI. More specifically, the first level of MDI is considered non-deprived, and all remaining levels (e.g., second, third, and fourth) of MDI are deprived. In the deprived level, the second, third, and fourth are the least, moderate, and high levels of deprivation, respectively. These levels of MDI, according to local governments with demarcation as non-deprived and deprived within seven provinces, are mentioned in Table 3.

Table 3 divulges that most of Nepal's local governments (f = 750, % = 99.6) score low MDI. Only three local governments accounted for non-deprivation. Two are in Bagmati, and the remaining one is in Lumbini province. Furthermore, two local governments of Karnali province were measured as highly deprived. Moreover, most local governments from Karnali, Madhesh, and Sudhur-Paschim provinces had a moderate level of disparity, accounting for 77.2%, 69.1%, and 62.5%, respectively. Beyond these, the remaining four provinces: Gandaki, Lumbini, Bagmati, and Koshi, hold 84.7%, 82.6%, 78.2%, and 65.4% of municipalities remark

**Table 3. Frequency of local governments carrying MDI values within provinces.**

| Province (N = 753) | Non-Deprived | Deprived | | | | Total |
|---|---|---|---|---|---|---|
| | | **Least** | **Moderate** | **High** | *Sub-total* | |
| Koshi | 0(0%) | 89(65.4%) | 48(35%) | 0(0%) | *137(100%)* | 137(100%) |
| Madhesh | 0(0%) | 42(30.9%) | 94(69.1%) | 0(0%) | *136(100%)* | 136(100%) |
| Bagmati | 2(1.7%) | 93(78.2%) | 24(20.2%) | 0(0%) | *117(98.3%)* | 119(100%) |
| Gandaki | 0(0%) | 72(84.7%) | 13(15.3%) | 0(0%) | *85(100%)* | 85(100%) |
| Lumbini | 1(0.9%) | 90(82.6%) | 18(16.5%) | 0(0%) | *108(99.1%)* | 109(100%) |
| Karnali | 0(0%) | 16(20.3%) | 61(77.2%) | 2(2.5%) | *79(100%)* | 79(100%) |
| Sudhur-paschim | 0(0%) | 33(37.5%) | 55(62.5%) | 0(0%) | *88(100%)* | 88(100%) |
| *Total* | *3(0.4%)* | *435(57.8%)* | *313(41.6%)* | *2(0.3%)* | *750(99.6%)* | *753(100%)* |

**Table 4. Disparity index of Nepal and its provinces according to its dimensions.**

| Province | ECODI | EDUDI | HDI | GCVI | LSDI | DDI |
|---|---|---|---|---|---|---|
| Koshi | 0.408 | 0.300 | 0.621 | 0.152 | 0.473 | 0.339 |
| Madhesh | 0.445 | 0.427 | 0.528 | 0.147 | 0.525 | 0.386 |
| Bagmati | 0.286 | 0.254 | 0.419 | 0.220 | 0.364 | 0.314 |
| Gandaki | 0.329 | 0.275 | 0.471 | 0.275 | 0.362 | 0.359 |
| Lumbini | 0.397 | 0.317 | 0.627 | 0.088 | 0.468 | 0.385 |
| Karnali | 0.483 | 0.331 | 0.696 | 0.280 | 0.556 | 0.426 |
| Sudhur-Paschim | 0.448 | 0.320 | 0.676 | 0.174 | 0.520 | 0.430 |
| *Overall, in Nepal* | *0.390* | *0.321* | *0.557* | *0.175* | *0.459* | *0.367* |

the least disparity, respectively. Overall, 435 (57.8%) and 313(41.6%) local governments of Nepal were accounted the least and moderate levels of disparity in their obtained MDI scores.

## Dimension-wise disparity index regarding provinces of Nepal

The summation of the Economy Disparity Index [ECODI], Educational Disparity Index [EDUDI], Health Disparity Index [HDI], Geography and Climatic Vulnerability Index [GCVI], Living Standard Disparity Index [LSDI], and Demography Disparity Index [DDI] contributes to determining MDI. The derived disparity index on six dimensions across provinces is presented in Table 4.

Karnali province scored the highest disparity value across all the dimensions except education and demography. Similarly, Madhesh consisted of high disparity (0.427) in education, and Sudhur-Paschim saw a high disparity (0.430) in the demographic dimension. Contrary to it, the Bagmati province accounts for the least disparity in most of the dimensions of the economy (0.286), education (0.254), health (0.419), and demography (0.314). Furthermore, this study also found the least value (0.088) in the geography and climatic vulnerability dimension in Lumbini province. Moreover, Gandaki province in the living standards dimension calculated (0.362) for the least disparity. Overall, health was noted the most and geography and climatic vulnerabilities the least discrepancy dimensions.

## MDI values of provinces

MDI values of Nepal and its provinces are derived from calculating multi-dimensional headcount ratios (H) and intensity of deprivations (A) presented in Table 5.

**Table 5. MDI score across the provinces.**

| Province | N | Ci(k) | Q | Ci(k)*n | H | A | MDI (H*A) |
|---|---|---|---|---|---|---|---|
| Koshi | 4534345 | 58.135 | 4534345 | 1758107 | 1 | 0.387 | 0.387 |
| Madhesh | 5404145 | 59.805 | 5404145 | 2263034 | 1 | 0.418 | 0.418 |
| Bagmati | 5529452 | 46.332 | 5529452 | 1794953 | 1 | 0.324 | 0.324 |
| Gandaki | 2403757 | 34.857 | 2403757 | 852957.3 | 1 | 0.354 | 0.354 |
| Lumbini | 4499272 | 45.404 | 4499272 | 1758750 | 1 | 0.390 | 0.390 |
| Karnali | 1570418 | 40.239 | 1570418 | 734049.6 | 1 | 0.467 | 0.467 |
| Sudhur-Paschim | 2552517 | 41.985 | 2552517 | 1119092 | 1 | 0.438 | 0.438 |
| Over all in Nepal | 26493906 | 326.76 | 26493906 | 10280943 | 1 | 0.388 | 0.388 |

*N = Population, Ci(k) = Censored Score, q = Population Disparity, H = Multi-dimensional Head Count Ratios, A = Intensity of Disparity

The study measured Karnali province as the most deprived province with the highest MDI value of 0.467, followed by Sudhur-Paschim at 0.438 and Madhesh at 0.418. In contrast, the least disparity is calculated (0.324) in Bagmati province. Gandaki (0.354) and Koshi (0.387) are measured as having the second and third least MDI values. Overall, the MDI value of Nepal is obtained at 0.388, which shows the nation with a multi-dimensional disparity.

## Discussions

This study elucidates that all the measured dimensions of MDI in all three tiers of government are not at a satisfactory level. The study accounts for HDI as the most and GCVI as the least deprived dimensions. Connecting to the findings, Nepal still has a 186 (per 100000 live births) Maternal Mortality Rate (MMR) [55] and 23 (per 1000 live births) Infant Mortality Rate (IMR) [56]. Both MMR and IMR are higher than those of its neighboring countries like China (MMR = 29 and IMR = 6) and Bhutan (MMR = 183 and IMR = 15) [56]. Similarly, the life expectancy at birth (years) of Nepal (68) is also lesser than in other neighboring countries like China (78) and Bhutan (72) [56]. Moreover, there are still no sufficient health facilities in remote areas and outside the Kathmandu Valley of Nepal [57] where most people are inhabitants [41]. These results also reveal that many people in Nepal are still deprived of quality health services and the lacking services in health further degrades their quality life [8].

The values of GCVI; however, accounted for the least among other dimensions. The finding is also analogous to the report of the Global Climate Risk Index (GCRI) 2021 [58], which puts Nepal and India in twelfth (GCRI = 20) and seventh rank (GCRI = 16.67) considering the GCRI for 2019. Nepal saw fewer fatalities than India and ranked tenth in fatalities in 2019 [58]. However, Nepal was accounted as one of the long-term climate risk countries with a CRI score of 31.33. These risks are estimated considering the effect of global warming that will melt the glaciers of the Himalayas and result in heavy rainfalls and landslides in highlands and floods in the plain area. Despite the earthquake in 2015 which hit Nepal and caused the loss of about 10,000 people, leaving several thousand injured [59], the occurrence of other incidences of natural calamities; however, seen less significantly in recent past [47]. Among the provinces, Gandaki province, nevertheless, faced noticeable calamities compared to other provinces. The province is mostly in hilly and mountainous areas, and it keeps the possibility of occurring many landslides and of heavy snowfall [60, 61].

The Bagmati province is the least deprived in HDI compared to other provinces. The province where the capital city is situated has mostly urban areas with concentrated health services. Most doctors and medical personnel are employed in the capital city [62]. For example, the Department of Health Services reports that Kathmandu valley has the highest number of hospitals [63]. Contrary to it, Karnali and Sudhur-Paschim scored the highest HDI. These provinces are also deprived of insufficient health facilities [63], people with the lowest life expectancy at birth [64], and the prevalence of high maternal and infant mortality rates [14]. This poor health status also puts these two provinces: Karnali and Sudhur-Paschim, in the most deprived rank of HDI than other provinces.

Besides the GCVI and LSDI, the Bagmati and Gandaki province has the least disparity on ECODI, EDUDI, and LSDI. The result shows that these two provinces have the least economic disparity. The report of the MOFAGA also shows these two provinces with the availability of infrastructures such as roads and airports, access to electricity and telecommunications, educational institutions and establishment of industry than other provinces [42].

The result also portrayed that Bagmati, Gandaki, and Koshi consisted of low DDI disparity than other provinces. This finding is analogous to the GII values of these provinces [11]. The least gap in gender, small size of families, more active population, and few households with a

disability are the resultants of the low DDI in these three provinces. In contrast, Karnali, Sudhur-Paschim, and Madhesh provinces are high in ECODI, EDUDI, LSDI, and DDI. These three provinces witness more disparity in the economy, education, living standard, and demography than the rest of the other provinces.

Bagmati and Gandaki provinces are found to have the least disparity with low MDI values. Contrary to it, MDI is measured highest in the Karnali province, consisting of highly deprived local governments and accounting for many local governments with disparity. This result, also analogous to the NPC and OPHDI, identified Bagmati as the least and Karnali province as the most deprived province of Nepal in multi-dimensional poverty [12]. The study by Chhetri and Malla also found Karnali as the deprived province with the least HDI and higher GII values [11]. These inequalities across the provinces have widened the socioeconomic and other forms of gap among the people across Nepal.

The elucidated disparity in each dimension (e.g., economy, health, education, living standard, demography, and climatic vulnerability) has formed the multi-dimensional disparity in Nepal. Such disparity is found to be associated with sociocultural, health, economic, geographical and ecological regions. These types of disparities within multi-sectors can be reduces by providing the equal access to services for the citizen [65]. Overall, this study found disparities at multiple dimensions; thus, the disparity at the multi-dimensional level is important to consider to achieve the targeted national goal of inclusive and equitable development in the federal context of Nepal.

## Conclusion

This study measured the disparity of Nepal in six dimensions composite of 34 indicators. The calculated MDI scores indicate that disparity prevails in all dimensions across all levels of the government. The MDI covered wide fields of dimensions and assessed many forms of disparity to portray the broader picture of disparity. There is an unequal distribution of access to the services and facilities to elevate the multi-dimensional nature of disparity. The distribution of the resources found centered in cities. More specifically, Bagmati province was relatively found in a better position while Karnali province accounted for having most disparity. The calculated score of local governments, with some exceptional, mostly falls under deprivation. This shows a big challenge in ensuring inclusive governance for all the citizens and institutionalizing the federal system in Nepal. So, these facts are considerable for policymakers to formulate policies and programs that reduce disparity. The local government, the smallest governance unit, could be a priority for intervention diminishing disparity as they are directly engaged in day-to-day citizens' activities. Province also needs important attention to reduce disparity. Although this study provides a robust approach to measuring disparity in multiple dimensions, several limitations must be highlighted. In most of the disparity indexes, households are taken as the unit of analysis; however, in this study, we considered the unit of analysis as local government.

In contrast to other studies, which normally assign values in binomial forms as 0 or 1; however, we put values of all indicators between 0 to 1. We relied on different secondary sources for data that were published in different periods. However, future researcher(s) may use first-hand sources and data of the same time frame. Similarly, this study's obtained results are equally applicable for forthcoming researchers as the reference of their studies and examine the contributions of ECODI, EDUDI, HDI, GCVI, LSDI, and DDI on MDI.

## Supporting information

**S1 Appendix.**
(DOCX)

## Author Contributions

**Conceptualization:** Prakash C. Bhattarai.

**Formal analysis:** Milan Shrestha.

**Methodology:** Prakash C. Bhattarai, Milan Shrestha, Prakash Kumar Paudel.

**Supervision:** Prakash C. Bhattarai.

**Writing – original draft:** Milan Shrestha, Prakash Kumar Paudel.

**Writing – review & editing:** Prakash C. Bhattarai, Prakash Kumar Paudel.

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
