## [Decision Letter · Decision Letter 0]

21 Feb 2023

PONE-D-22-27369Multi-Dimensional Disparity Index (MDI) and its Dimensions: A Case of  NepalPLOS ONE

Dear Dr. Bhattarai,

Thank you for submitting your manuscript to PLOS ONE. After careful consideration, we feel that it has merit but does not fully meet PLOS ONE’s publication criteria as it currently stands. Therefore, we invite you to submit a revised version of the manuscript that addresses the points raised during the review process.

Authors need to incorporate all changes suggested by editor and reviewers, 

We look forward to receiving your revised manuscript.

Kind regards,

Muhammad Tayyab Sohail

Academic Editor

PLOS ONE

2.We note that you have indicated that data from this study are available upon request. PLOS only allows data to be available upon request if there are legal or ethical restrictions on sharing data publicly. For more information on unacceptable data access restrictions, please see http://journals.plos.org/plosone/s/data-availability#loc-unacceptable-data-access-restrictions.

3.Please include your tables as part of your main manuscript and remove the individual files. Please note that supplementary tables (should remain/ be uploaded) as separate ""supporting information"" files.

Additional Editor Comments:

To further improve the text, I suggest the following changes in the manuscript.

Abstract: Abstract should be written in concise. I would suggest listing only some of the most important results to justify the implications and conclusions of the study.

The background of an introduction should be revised accordingly.

The introduction is very good. It doesn't reflect the goal; please rewrite it again, it is suggested to include some latest reference.

Objectives of this study must be included at end of introduction part.

I highly recommended to authors, if possible, please modify the figure with good quality images.

The economic intuition behind the results are missing. The author/s should revise the discussion part. The result should be supported with recent studies.

What is contribution of this work to existing literature?

It has been observed that the authors have used old references and ignored the latest studies. So it is suggested to add recent references. Please check reference section some references are missing.The policy implications also required elaboration. The implications should go along with the results and the course of action should be discussed in this part. In some places, some grammatical errors are found that need to be fixed.

Reviewers' comments:

Reviewer's Responses to Questions

**Comments to the Author**

1. Is the manuscript technically sound, and do the data support the conclusions?

Reviewer #1: Yes

Reviewer #2: Partly

2. Has the statistical analysis been performed appropriately and rigorously? 

Reviewer #1: No

Reviewer #2: Yes

3. Have the authors made all data underlying the findings in their manuscript fully available?

Reviewer #1: No

Reviewer #2: Yes

4. Is the manuscript presented in an intelligible fashion and written in standard English?

Reviewer #1: Yes

Reviewer #2: No

5. Review Comments to the Author

Reviewer #1: This manuscript has been selected from a relatively novel perspective. The comments as follows.

1.The following points should be clearly stated in the introduction：

What are the limitations of the MDI index, and what are the advantages of the index for Nepal? Are there any limitations for Nepal?

In the introduction, the purpose of the study and the significance of the study should be further clarified.

2.The number of government officials, international NGO workers, MDI experts, economists, academicians, announcement health experts and sociologists involved in determining MDI dimensions and indicators in the research methodology. And their reliability and validity in this process, such as t-test.

3.It is suggested that the basic profiles of the seven provincial governments of Nepal, which are the subject of the study, should be introduced in the research methodology including physical geography, climate, economy, culture, population, politics, education, etc.

4. The results are only a relatively simple description of Table 2, without deeper excavation, not to mention the analysis of the actual situation in the provinces and cities.

5.The discussion does not provide a good argument for the implementability of MDI index in Nepalese applications and its scientific validity.

6.The discussion should further analyze and draw out the particularities and commonalities of Nepal and its provinces in terms of economy, education, health, geography, population, and life, and the reasons for these individualities and commonalities.

After revising the above items, I recommend accepting this manuscript.

Reviewer #2: Research title may be revised. Most of the he references / citations mentioned in the article are very old the research objectives and rationale required to be properly specified. Research design is weak. article proofreading is highly recommended. The study period to calculate MDI not mentioned to assess or make conclusions.

6. PLOS authors have the option to publish the peer review history of their article (what does this mean?). If published, this will include your full peer review and any attached files.

Reviewer #1: No

Reviewer #2: No

---

## [Author Response · Author response to Decision Letter 0]

15 Apr 2023

It has been submitted in a separate file.

---

## [Decision Letter · Decision Letter 1]

11 May 2023

Measuring Multi-dimensional Disparity Index: A Case of Nepal

PONE-D-22-27369R1

Dear Dr. Bhattarai,

We’re pleased to inform you that your manuscript has been judged scientifically suitable for publication and will be formally accepted for publication once it meets all outstanding technical requirements.

Kind regards,

Muhammad Tayyab Sohail

Academic Editor

PLOS ONE

Additional Editor Comments (optional):

Reviewers' comments:

Reviewer's Responses to Questions

**Comments to the Author**

1. If the authors have adequately addressed your comments raised in a previous round of review and you feel that this manuscript is now acceptable for publication, you may indicate that here to bypass the “Comments to the Author” section, enter your conflict of interest statement in the “Confidential to Editor” section, and submit your "Accept" recommendation.

Reviewer #1: All comments have been addressed

2. Is the manuscript technically sound, and do the data support the conclusions?

Reviewer #1: Yes

3. Has the statistical analysis been performed appropriately and rigorously? 

Reviewer #1: Yes

4. Have the authors made all data underlying the findings in their manuscript fully available?

Reviewer #1: (No Response)

5. Is the manuscript presented in an intelligible fashion and written in standard English?

Reviewer #1: Yes

6. Review Comments to the Author

Reviewer #1: This article comprehensively evaluates the differences between seven provinces in Nepal through Multi-dimensional Disparity Index (MDI). I have seen the great efforts made by the authors, resulting in greatly improved content. Here are some minor suggestions for this article, be specific:

1. 1-2 sentences should be added to the abstract to briefly introduce the purpose of this research. For example, the contribution and use of Multi-dimensional Disparity Index (MDI).

2. Keywords are of substantial significance in expressing the central content of the thesis. I think the current 7 keywords are a bit redundant. Therefore, I suggest replacing it with more concise and generalized vocabulary, which should be able to clearly and intuitively express the topic of literature discussion or expression.

3. What should the authors further consider to be the contribution of this study to the country or region? Therefore, I suggest that the author put forward some relevant policy recommendations based on the results of the current research.

7. PLOS authors have the option to publish the peer review history of their article (what does this mean?). If published, this will include your full peer review and any attached files.

Reviewer #1: No

---

## [Editor Report · Acceptance letter]

17 May 2023

PONE-D-22-27369R1 

Measuring Multi-dimensional Disparity Index: A Case of Nepal 

Dear Dr. Bhattarai:

I'm pleased to inform you that your manuscript has been deemed suitable for publication in PLOS ONE. Congratulations! Your manuscript is now with our production department. 

Kind regards, 

on behalf of

Dr. Muhammad Tayyab Sohail 

Academic Editor

PLOS ONE